# In Vitro Biodegradation of a-C:H:SiO_x_ Films on Ti-6Al-4V Alloy

**DOI:** 10.3390/ma15124239

**Published:** 2022-06-15

**Authors:** Alexander Grenadyorov, Andrey Solovyev, Konstantin Oskomov, Ekaterina Porokhova, Konstantin Brazovskii, Anna Gorokhova, Temur Nasibov, Larisa Litvinova, Igor Khlusov

**Affiliations:** 1The Institute of High Current Electronics SB RAS, 2/3, Akademichesky Ave., 634055 Tomsk, Russia; 1711sasha@mail.ru (A.G.); andrewsol@mail.ru (A.S.); oskomov@yandex.ru (K.O.); 2Department of Morphology and General Pathology, Siberian State Medical University, 2, Moskovskii Trakt, 634050 Tomsk, Russia; porohova_e@mail.ru (E.P.); a.gorokhova3062@gmail.com (A.G.); temur.nsbv@gmail.com (T.N.); 3Research School of Chemistry and Applied Biomedical Sciences, National Research Tomsk Polytechnic University, 43-A, Lenin Ave., 634050 Tomsk, Russia; bks_2005@mail.ru; 4Center for Immunology and Cellular Biotechnology, Immanuel Kant Baltic Federal University, 14A, Nevskii Str., 236016 Kaliningrad, Russia; larisalitvinova@yandex.ru

**Keywords:** diamond-like nanocomposite, a-C:H:SiO_x_ film, adhesion, five-week biodegradation, 0.9% NaCl solution

## Abstract

This paper focuses mainly on the in vitro study of a five-week biodegradation of a-C:H:SiO_x_ films of different thickness, obtained by plasma-assisted chemical vapor deposition onto Ti-6Al-4V alloy substrate using its pulsed bipolar biasing. In vitro immersion of a-C:H:SiO_x_ films in a solution of 0.9% NaCl was used. It is shown how the a-C:H:SiO_x_ film thickness (0.5–3 µm) affects the surface morphology, adhesive strength, and Na^+^ and Cl^−^ precipitation on the film surface from the NaCl solution. With increasing film thickness, the roughness indices are reducing a little. The adhesive strength of the a-C:H:SiO_x_ films to metal substrate corresponds to quality HF1 (0.5 µm in thickness) and HF2-HF3 (1.5–3 µm in thickness) of the Rockwell hardness test (VDI 3198) that defines strong interfacial adhesion and is usually applied in practice. The morphometric analysis of the film surface shows that on a-C:H:SiO_x_-coated Ti-6Al-4V alloy surface, the area occupied by the grains of sodium chloride is lower than on the uncoated surface. The reduction in the ion precipitation from 0.9% NaCl onto the film surface depended on the elemental composition of the surface layer conditioned by the thickness growth of the a-C:H:SiO_x_ film. Based on the results of energy dispersive X-ray spectroscopy, the multiple regression equations are suggested to explain the effect of the elemental composition of the a-C:H:SiO_x_ film on the decreased Na^+^ and Cl^−^ precipitation. As a result, the a-C:H:SiO_x_ films successfully combine good adhesion strength and rare ion precipitation and thus are rather promising for medical applications on cardiovascular stents and/or friction parts of heart pumps.

## 1. Introduction

Ti-6Al-4V alloy (grade 5) is the most medically used among titanium (Ti)-based alloys [1]. However, it can release toxic vanadium ions during a corrosion caused by biological fluids [2]. Therefore, the thin film deposition is the most widely used method of the surface modification, which allows also to easily vary and regulate the surface material properties for a specific applications. Diamond-like carbon (DLC) materials provide the surface with improved mechanical [3], tribological [4], and anticorrosion properties [5]. Among such materials are DLC films alloyed by Si or SiO_x_, known as DLC:Si, DLC:SiO_x_ or a-C:H:SiO_x_. Si-alloyed DLC films are characterized by low (<1 GPa) internal stress, relatively high (10–20 GPa) hardness, low (<0.1) friction coefficient, wear rate (~10^−7^ mm^3^/N·m) [6], corrosion rate (10^−6^ mm/year) [7], and rather interesting medical and biological properties such as biocompatibility and non-thrombogenicity [8,9].

Lately, much research on the structure and properties of the Si/SiO_x_-incorporated DLC films has been conducted [6,8,10,11,12,13]. In the last decade, the impact of incorporated silicon on the mechanical [14,15], tribological [12,16], anticorrosion [17,18,19], and medical and biological [10,20,21] properties of DLC films has attracted much attention from research teams. It is found that the silicon addition has a positive effect on the final properties of thin films. According to [12,18], the optimum Si concentration provides the lower friction coefficient and, consequently, the lower corrosion rate [17,18]. Additionally, the Si incorporation assists in the lower activity of inflammatory reactions, adhesion of platelets, and improves the biocompatibility of medical devices without regard for the film fabrication technique [9,10,22,23].

A study of the thin film behavior in electrolytes is largely determined by their prospective use as a smooth coating of the blood-contacting friction components of mechanical heart pumps and cardiovascular stents. At the same time, Si incorporation can increase film solubility that was demonstrated for calcium phosphate coatings [24]. On the other hand, a possibility of biomineralization of thin Si/SiO_x_-incorporated DLC films [25] conditioned by protein and ion precipitation from biological fluids on the surfaces poses risks to their functioning. At least, Si/SiO_x_-DLC coatings may increase surface roughness and in vitro adhesion of albumin [26]. There is, however, no information in the literature about biodegradation of a-C:H:SiO_x_ films and the influence of their thickness on the resistance to body fluids. Teske et al. [27] performed accelerated degradation study of aluminum samples with plasma polymer coatings in 1 M hydrochloric acid for about 2 h. In the present paper, to obtain more relevant information on the biodegradation of a-C:H:SiOx films, the last were immersed in 0.9% NaCl solution at 37 °C for 5 weeks.

Thus, this paper mainly focuses on the in vitro study of a five-week biodegradation of a-C:H:SiO_x_ films of different thickness, obtained by plasma-assisted chemical vapor deposition onto Ti-6Al-4V alloy substrate using its pulsed bipolar biasing.

## 2. Materials and Methods

### 2.1. Preparation of Ti-6Al-4V Alloy Substrates

The Ti-6Al-4V (Grade 5) titanium sheet (VSMPO-AVISMA Corporation, Verkhnyaya Salda, Russia) was used to fabricate the test substrates due to its widespread use in medicine [1]. The results of the elemental analysis of the titanium alloy are presented in Table 1. Rounded substrates of diameter 10 mm and 1 mm thickness were cut using the electrical discharge machining. Their surface was sanded by sandpaper 2000 grit and then cleaned ultrasonically in isopropyl alcohol and acetone for 10 min in each liquid.

### 2.2. Preparation of a-C:H:SiO_x_ Coatings

The plasma-assisted chemical vapor deposition (PACVD) described in [28,29,30] was used to obtain a-C:H:SiO_x_ films. The PACVD process was provided by the plasma generator with hot cathode. Polyphenyl methylsiloxane (PPMS) was used as a precursor; its chemical formula is (CH_3_)_3_SiO(CH_3_C_6_H_5_SiO)_3_Si(CH_3_)_3_.

The general configuration of the PACVD vacuum system is illustrated in Figure 1. When heated by the hot cathode, liquid PPMS supplied via a metal capillary, transforms to the vapor phase. The PPMS molecules are subjected to dissociation a partial ionization. The obtained plasma is used to generate a-C:H:SiO_x_ films. The property control of these films can be provided through varying the deposition parameters such as process pressure, precursor consumption, discharge current, filament current, and substrate bias voltage. Our previous research [28,29,30] reports on the effect from these parameters on different properties of the a-C:H:SiO_x_ film.

In our experiment, the a-C:H:SiO_x_ film deposition is performed in optimum conditions, which provide the formation of hard, wear-resistant, and biocompatible coatings [31,32]. In PACVD, the process pressure is 0.1 Pa at the argon gas rate of 4 ± 0.1 L/h and PPMS consumption rate of 0.8 ± 0.05 mL/h. Other process parameters include a constant 9 ± 1 V voltage applied to the filament, 50 ± 4 A filament current, 140 ± 5 V arcing voltage, 6 ± 0.5 A discharge current, 300 ± 30 V bipolar pulsed bias voltage with the negative pulse amplitude applied to the substrate holder, 100 kHz pulse repetition frequency, and 60% duty cycle. During the ion-plasma surface cleaning for 10 min, the negative pulse amplitude of the substrate bias voltage is 1000 ± 50 V and the gas pressure is 0.3 Pa without the use of PPMS.

### 2.3. Surface Characterization

The a-C:H:SiO_x_ coating thickness was studied by Calotest technique using a CAT-S-0000 Calotest machine (CSEM, Neuchâtel, Switzerland) with a steel ball of diameter 35 mm with 1 µm abrasion by diamond particles. The FEI Quanta 200 3D dual beam system (FEI Company, Hillsboro, OR, USA) combining a scanning electron microscope coupled with an energy dispersive X-ray analyzer (SEM/EDX) and a focused ion beam microscope was used to characterize the morphology and microstructure of thin films. The accelerating voltage was 20 kV. The elemental composition of the obtained films was examined in the area of 350 × 250 µm. For the statistical summary report, the elemental composition was analyzed at least in 5 randomly sampled regions. The surface morphology and roughness were studied in a 15 × 15 µm area using a Solver P47 scanning probe microscope (SPM) (NT-MDT, Moscow, Russia) in a tapping mode. The surface defects of as-deposited a-C:H:SiO_x_ films and the Rockwell hardness were examined on a Polar-1 metallographic microscope (Mikromed, Moscow, Russia) in the reflected polarized light.

### 2.4. Adhesive Strength

Rockwell C scale hardness (HRC) test (the VDI 3198 indentation test [33]) was conducted for the adhesive strength evaluation of a-C:H:SiO_x_ films. A TK-2 spheroconical diamond indenter (IVMASHPROM, Yekaterinburg, Russia) with a 120° included cone angle was used for nanoindentation examinations. The preliminary and full indentation load was 10 kg (~100 N) and 150 kg (1471 N), respectively. The testing cycle continued for 4 s. After testing, the film adhesion to its substrate ranged from perfect (HF1) to poor adhesion (HF6), depending on the number of cracks and delamination (see Figure 2). The peeling coefficient is equal to the ratio of the area of coating peeling to the area of the hole left by the indenter. The adhesion of the coating is better when the value of the peeling coefficient is smaller.

### 2.5. Substrate Sterilization for Biodegradation Testing

Prior to the in vitro testing, the substrates were placed in sterile plastic tubes for disinfection in a 3% H_2_O_2_ solution at 37 °C for 180 min and then dried in a biological safety laminar flow cabinet (AMS-MFME, Miass, Russia) to prevent them from contamination. The sterilization procedure was provided by a 100% ethylene oxide vapor in a GS Series sterilizer/aerator Steri-Vac (3M, Maplewood, MN, USA), as described in [34]. No changes in the surface properties or substrate weight were found after sterilization.

### 2.6. In Vitro Biodegradation

In accordance with the ISO 10993-15:2019, the substrate biodegradation was studied by their immersion in a 0.9% NaCl solution (OOO “Mosfarm”, Bogorodskoe, Russia). This solution was selected due to physiological Na^+^ and Cl^−^ concentrations and the absence of micro-impurities as compared to natural and synthetic biological media. The substrates were cultured at 37 °C for 5 weeks (2 mL of NaCl solution per a substrate in accordance with the ISO 10993-12:2007), as described in [35]. These conditions meet the ISO 10993-9:2009 requirements for medical implants being over 30 days in organism. A sterile 15 mL polypropylene conical tube with a screw cap was used for each substrate to immerse into a 0.9% NaCl solution. At the end of each week, the substrate extracts were fully replaced with a fresh portion of NaCl solution. The NaCl solution without test substrates was used as a reference.

After 5-week immersion, the substrates were removed from tubes and dried in ambient conditions for 3 weeks and then weighted on a GR-202 Semi-Micro Analytical Balance (A&D Company, Tokyo, Japan). The mass change was calculated before and after the immersion. Five substrates from each group were tested in triplicate. The extracts were collected and kept at −80 °C.

The Na^+^ and Cl^−^ concentrations in the solutions were determined by ion-selective electrode sets from Thermo Fisher Scientific (Waltham, MA, USA) using a Konelab 60i automatic biochemical analyzer (Thermo Fisher Scientific, USA). All procedures were performed as recommended by the manufacturer. The measurement error does not exceed 1%.

### 2.7. Sodium Chloride Precipitation

The SEM/EDX system with the INCA analyzer (Oxford Instruments, Abingdon, UK) was used to investigate the elemental composition and distribution over the substrate surface. Computer-assisted morphometry was performed for quantitative measurement of NaCl deposition. The ImageJ 1.38 image processing program (National Institutes of Health, Bethesda, MD, USA) facilitated the morphometric analysis to calculate the areas of the NaCl precipitates on the substrate surfaces.

### 2.8. Statistical Analysis

Statistical analyses were performed using the STATISTICA 13.3 software package (TIBCO Software Inc., Palo Alto, CA, USA) for Windows. The data are shown as the mean (X), standard deviation (SD), and standard error (SE) for physical results and 25% lower quartile (*Q*1), median (*Me*) and 75% upper quartile (*Q*3). The distribution normality was defined by the Shapiro–Wilk test. If chemical results were not normally distributed, the non-parametric Mann–Whitney U test (*P*_U_) and Wilcoxon signed-rank test (*P*_T_) were conducted to evaluate a significant difference between the independent and dependent substrates, respectively. Otherwise, Student’s *t*-test was conducted. Spearman’s rank correlation coefficient (*r*_S_), dual regression (*r*) and multiple linear regression analyses with an *F*-test were used in the experiments. Statistically significant differences were considered at *p*-value < 0.05.

## 3. Results

### 3.1. Surface Morphology

SEM images of the Ti-6Al-4V alloy surface coated with the a-C:H:SiO_x_ film of different thickness are given in Figure 3. In general, the a-C:H:SiO_x_ film surface replicates the substrate texture. At the film thickness of 0.5 µm, the surface has grooves originating from sanding the initial substrate surface. The increase in the film thickness makes the surface smoother. The grain structure is observed on the film surface 3 µm thick.

The SPM images of the surface morphology in the area 15 × 15 µm are presented in Figure 4. One can see the smooth surface texture with increasing film thickness. When the film thickness is relatively thin, ~500 nm, the texture is rough, as the grooves and scratches produced by abrasive machining are deeper than the film thickness. The film thickness of 1500 nm makes these damages less visible. The 3000 nm thick film eliminates them; only the film texture with peaks and valleys is observed. With increasing film thickness, the root-mean-square roughness *R*_q_ reduces from 51 to 36 nm. The surface roughness values are summarized in Table 2.

### 3.2. Adhesion of a-C:H:SiO_x_ Films

The Rockwell hardness test (VDI 3198) is often used to evaluate the adhesion of hard coatings to various materials [36,37,38,39,40]. In conformity with the VDI 3198, the adhesive strength quality HF1–HF4 defines strong interfacial adhesion and is usually applied in practice, whereas HF5 and HF6 define poor adhesion and are not acceptable [33]. We use this evaluation method to study the a-C:H:SiO_x_ film adhesion depending on its thickness.

In Figure 5, the optical micrographs demonstrate the Ti-6Al-4V surface with deposited a-C:H:SiO_x_ films. At a 0.5 µm film thickness, the indentation craters formed after the Rockwell hardness test have no spalling and laminations. It means that the film adhesion quality is very high (HF1). At 1.5 and 3 µm film thickness, the crater edges are slightly spalled and delaminated. The adhesive strength quality of these films is HF2–HF3, and they can be used further. The penetration in a relatively soft substrate surface is rather deep (~150 µm in our experiment), and plastic deformation causes the displacement of the substrate material, which induces spallation and microcracks of the crater edges even in hard coatings. To accomplish a smooth transition between the hard and soft substrates to alleviate the eggshell-like effect, the surface is preliminarily hardened for the improvement of its bearing capacity [37].

### 3.3. Visual Examination and Weighting of Substrates after Biodegradation

Visual examination of substrates after a five-week in vitro biodegradation did not show changes in their geometry (see Appendix A) as compared to their initial state. The coatings were not damaged. The metallic appearance observed at the edges of substrates 2-1, 2-5, 3-2, 3-5 was caused by retainer plates used in film depositing.

In Appendix A, one can see surface defects on the a-C:H:SiO_x_ film 3 µm thick. The substrate 4-2 has a rippled surface; the ripples have different shape and size and are separated by the light lines. Light areas are observed at the bottom of the substrates 4-3 and 4-5.

Thus, after a five-week in vitro biodegradation of the substrates in isotonic NaCl solution, the a-C:H:SiO_x_ films demonstrate visual changes, especially in films 3 µm thick. These changes are probably caused by the ion precipitation on the substrate surface from the NaCl solution.

According to the results of the five-week biodegradation presented in Table 3, the weight of Ti-6Al-4V bare substrates remained unchanged, which proved their low biodegradation during long-term immersion in weak electrolyte used as a body fluid or blood.

The weight of a-C:H:SiO_x_-coated substrates is some 0.02–0.03% higher, when the film thickness grows compared to the respective values before biodegradation. The difference does not, however, reach the statistical values (see Table 3).

The phenomenon of inverse salt precipitation from the solution onto an artificial surface was earlier observed in [41] in the experiment with oxynitride films. However, in the case of the a-C:H:SiO_x_ film, the ion precipitation process was rather delicate, as the accurate weighting did not allow to evaluate it. In this connection, we measured Na^+^ and Cl^−^ concentration in the extracts to identify their possible changes after 5 weeks of in vitro biodegradation.

### 3.4. Identification of Na^+^ and Cl^−^ Concentration during 5 Weeks of In Vitro Biodegradation

In general, the data given in Appendix A show the ion deviation of not over 1.5–2.7% of the reference solution in the extracts. Thus, Ti-6Al-4V substrates demonstrate a stable behavior of the solid-solution system during the experiment.

The a-C:H:SiO_x_ thin films did not considerably affect Na^+^ and Cl^−^ concentration in extracts if compared to the uncoated Ti-6Al-4V substrates. The exception was the fourth week, when Na^+^ concentration in the 1.5 µm thick film was 0.9 mmol higher than in the bare metal substrate (*P*_U_ < 0.05). Against the solver without substrates, Na^+^ and Cl^−^ concentration in extracts occasionally increased in the whole thickness range of 0.5 to 3 µm, as presented in Appendix A.

The obtained data reject the hypothesis about a significant salt precipitation on the a-C:H:SiO_x_ film. A wavelike salt precipitation/dissolution probably can occur on the substrates due to a weekly replacement of the solvent portions. Moreover, a significant area of the vial surface can contribute to fluctuations in Na^+^ and Cl^−^ concentration in the extracts.

Therefore, the elemental analysis of NaCl precipitates was conducted and the precipitation area was measured after the fifth week of in vitro biodegradation.

### 3.5. Results of SEM, EDX and Computer-Assisted Morphometry of Precipitates

According to SEM observations in Figure 6, the grains of sodium chloride precipitate on the substrate surface after the five-week immersion in the NaCl solution. The grain shape is typical for sodium chloride. Therefore, the EDX analysis is conducted at several points and areas of the substrate surface. Moreover, one can see the difference in the precipitation area in different substrates. We thus calculate the total area occupied by these grains per surface area in SEM images (see Table 4).

Point measurements of cubic NaCl grains on a diverse surface of substrates demonstrate almost the same median ratio Na/Cl_wt.%_ = 0.665, which approaches to the reference coefficient 0.648 based on the atomic weight of Na (23 a.e.) and chlorine (35.46 a.e.)

According to Figure 6b, a significant amount of NaCl precipitates is observed on the uncoated Ti-6Al-4V substrate surface. Additionally, according to Table 5, this is 9.63 at.% for Na and 4.15 at.% for Cl ions, as the EDX analysis shows. The a-C:H:SiO_x_ film thickness of 0.5 to 3 µm provides a statistically significant decrease by 97–99.5% for Na^+^ precipitation and by 96–98.5% for Cl^−^ precipitation.

These surface changes caused by its interaction with the NaCl solution, are accompanied by the regression dependence between the carbon and silicon content in the a-C:H:SiO_x_-coated substrate surface. This is shown in Figure 7.

Spearman’s rank correlation shows a strong invert dependence (*r*_S_ = 0.73–0.75, *p* < 0.001, *n* = 40) between Na^+^, but not Cl^−^ decreasing precipitation with increasing carbon and silicon ion concentration in the surface layer ~3 µm thick. The EDX measurements show that the change in the chemical composition of the surface layer results from the protective effect of the a-C:H:SiO_x_ film, which prevents the ion precipitation. At the same time, the direct correlations for decreasing titanium and oxygen levels with falling Na^+^ precipitation are medium (*r*_S_ = 0.63–0.70, *p* < 0.001, *n* = 40), while for chlorine, it is not identified (*r*_S_ = 0.3–0.31, *p* > 0.05, *n* = 40).

The computer-assisted morphometry of the a-C:H:SiO_x_-coated substrate surface after five-week biodegradation demonstrates a 5–20-fold decrease in the area occupied by grains compared to the uncoated substrates (see Table 4 and Figure 6). At the same time, Spearman’s rank inverse correlation (*r*_S_ = −0.76, *p* < 0.001, *n* = 40) is detected for the grain-occupied area and the a-C:H:SiO_x_ film thickness on the substrate surface.

### 3.6. Multiple Linear Regression

According to Spearman’s rank correlation, the C and Si concentration (*r*_S_ = −0.96, *p* < 0.001, *n* = 40) grows in the Ti-6Al-4V substrate surface, whereas the Ti (*r*_S_ = −0.96, *p* < 0.001, *n* = 40) and O (*r*_S_ = −0.87, *p* < 0.001, *n* = 40) concentration lowers with increasing a-C:H:SiO_x_ film thickness (see Table 5). It raises the question about the potential contribution of the chemical elements of the C:H:SiO_x_ film deposited onto the Ti-6Al-4V substrate to the ion precipitation from the solvent.

The formula of the multiple linear regression of the ion concentrations (at.%) is
[Na] = −0.63193[C] − 0.79058[O] − 0.53708[Si] − 0.59384[Ti](1)

With the determination coefficient *R*^2^ = 0.9936, *p* < 0.001 in conformity with an *F*-test.

The regression coefficients in Table 6, prove a close (*p* < 0.001) linear relationship between Na^+^ precipitation and the growing content of carbon, oxygen, silicon, and titanium in the substrate surface. This relationship can be expressed as O > C > Ti > Si.

According to Table 5, the oxygen and titanium concentration reduces by 2 and over 50 times at a 3 µm film thickness, respectively. It follows from (1) that the contribution of these elements to the reduction in Na^+^ precipitation is 24.5 and 38 times lower than that of carbon and silicon, respectively, as compared to the uncoated titanium substrate.

Almost the same situation is observed for Cl^−^ precipitation, but with the lower values of the multiple regression (*R*^2^ = 0.9861, *p* < 0.001 according to *F*-test):[Cl] = −0.30606[C] − 0.31833[O] − 0.30112[Si] − 0.30574[Ti](2)

As can be seen from Table 6, the real effect (*p* < 0.001) from these elements in the substrate surface on regression coefficients of Cl^−^ precipitation is almost the same. Nevertheless, their total contribution to the reduction in Cl^−^ precipitation on the substrate surface is largely the same as to Na^+^ precipitation.

## 4. Discussion

DLC films, including Si-alloyed DLC films, possess a high hemo- and biocompatibility [42]. However, long-term contact with the blood and its components can cause the surface mineralization of medical devices [43,44], which is a mechanical threat for friction components of mechanical heart pumps.

Na^+^ and Cl^−^ are the main blood electrolytes. The chlorothiazide sodium stability in in vitro contact with medical devices may change after six-day storage [45]. However, no data about behavior of a-C:H:SiO_x_ films and the influence of their thickness on the resistance to body fluids were found in the literature. In our experiment, the five-week biodegradation is conditioned by the expected application of the substrates as implantable stents or parts of heart pumps for 30 days and longer, as well as the presence of relatively bioinert Ti-6Al-4V substrates and carbon coatings [46].

It is considered that the protein adsorption from the blood and tissue fluid is a determinant for the hemocompatibility of medical devices [47]. On the other hand, the isotonic (0.9%) NaCl solution, the most widely employed in hematology and transfusion medicine [48], is recommended by the international standards of corrosion testing and investigations of biodegradation of medical materials.

Isotonic NaCl solution usually facilitates the release of elements from a solid body (coating). Nevertheless, an inverse salt precipitation from the solution onto thin films is also observed after the five-week in vitro precipitation [41].

The degradation products were evaluated in accordance with the ISO 10993-9 (ISO 10993-9:2019), namely “Biological evaluation of medical devices—Part 9: Framework for identification and quantification of potential degradation products”, including changes in the weight and appearance of the material, quantification of degradation products based on the appropriate analytical approaches.

In general, DLC films protect metal surfaces from electrolytic corrosion [19]. At the same time, the protective effect from Si-alloyed DLC films can be considerably lower than from pure DLC films [25]. The biological behavior of DLC films largely depends on the deposition technique [42]. At least 74 types of amorphous DLC films are currently classified [49].

In our research, PACVD of the a-C:H:SiO_x_ film onto the AISI 316L stainless steel significantly reduces their corrosion rate in a solution of 0.5 M NaCl [50]. A more significant reduction in the corrosion rate is observed for the a-C:H:SiO_x_ film deposited onto the Ti-6Al-4V alloy surface, when exposed to 0.5 M NaCl and PBS binding buffer at 22 and 37 °C, respectively [31]. In the experiment with Ti-6Al-4V substrates, their five-week in vitro biodegradation in a solution of 0.9% NaCl is accompanied by the salt precipitation (see Appendix A) onto the surface (see Figure 7 and Table 5). On the contrary, the increase in the C and Si content in the surface layer with increasing thickness of the a-C:H:SiO_x_ film, unambiguously retards the salt precipitation, as shown in Table 5. This is accompanied by a reduction in the area occupied by the precipitated grains (Table 4).

Mechanisms of the deposited a-C:H:SiO_x_ film effect are still unclear. On the one hand, the beneficial effect of the Si admixture is attributed to the improvement of the surface roughness showing slightly lower average and root-mean-square roughness indices *R*_a_ and *R*_q_ [19]. According to Table 2, the root-mean-square roughness *R*_q_ reduces with increasing a-C:H:SiO_x_ film thickness up to 3 µm. However, Randeniya et al. [20] report that the average film roughness increases while Si and SiO_x_ are added to DLC films.

Early research [41] documented that zeta potential of thin films and further ion precipitation were affected by the electrostatic surface charge. In [31], we proved that a-C:H:SiO_x_ films possessed the negative surface potential in air; its amplitude grew with elevating thickness of the a-C:H:SiO_x_ coating on titanium, silicon, and steel substrates [50]. At the same time, Spearman’s rank correlation showed that the growth in the a-C:H:SiO_x_ film thickness prevented Na^+^ precipitation (*r*_S_ = −0.73–0.75, *p* < 0.001, *n* = 40) rather than Cl^−^ precipitation (*r*_S_ = −0.30–0.33, *p* > 0.05, *n* = 40). This was illogical with respect to the negative sign of the electrostatic potential of the a-C:H:SiO_x_ film. A non-uniform distribution of the electrostatic potential over the film surface might have utmost significance; its amplitude varied from −133 to 227 mV at its average value of −28 mV [32].

Moreover, according to the EDX analysis, the increase in the film thickness on the Ti-6Al-4V alloy substrate after the five-week immersion in 0.9% NaCl, is accompanied by the linear growth in the C and Si concentration (see Figure 7) and concurrent decrease in the O concentration (*r* = −0.79, *p* < 0.001, *n* = 40; *y* = 6.62 − 1.25*x*). In turn, the lower O concentration in the coating directly correlates with the low Na^+^ precipitation (*r*_S_ = 0.66, *p* < 0.001, *n* = 40), rather than Cl^−^ precipitation (*r*_S_ = 0.34, *p* > 0.05, *n* = 40). This is probably because the oxide content in the a-C:H:SiO_x_ film after a five-week biodegradation in 0.9% NaCl can affect its phase composition and, consequently, the precipitation of sodium chloride on the substrate surface.

The multiple linear regression proves that Na^+^ and Cl^−^ precipitation from the solution onto the a-C:H:SiO_x_ film depends on the content of not only chemical elements, but also oxides. Actually, the multiple regression equation confirms the negative relation between Na^+^ and Cl^−^ (in a less degree) concentration on the a-C:H:SiO_x_ film surface and the growth in the C and Si content in its composition. The negative multiple regression of sodium/oxygen with the reduction in their content with increasing film thickness (Table 5) contradicts with their pair positive correlation.

Therefore, the reduction in the ion precipitation from 0.9% NaCl onto the film surface depends on the elemental composition of the surface layer conditioned by the thickness growth of the a-C:H:SiO_x_ film.

Cardiovascular diseases (CVDs) are widespread all the world. Blood contacting stents and the components of extracorporeal blood circulation (e.g., heart pumps) are one of the leading devices for CVD treatment [15]. All the test a-C:H:SiO_x_ films with 0.5-3 µm thicknesses dramatically reduced Na^+^ and Cl^−^ precipitation (Table 5) and their crystallite deposition (Table 4) on the substrate surface compared to uncoated Ti-6Al-4V alloy. The fattest film showed the best decrease in Na^+^ deposition (Table 5); however, its adhesive strength quality HF2–HF3 was lower compared to 0.5 µm thick film (Figure 2). Coating spalling and microcracking hold real mechanical hazard for a lifetime of stent surface and friction components of mechanical heart pumps. Hence, an identification of the a-C:H:SiO_x_ thickness with optimal chemical and mechanical features is of great interest for future research and possible biomedical application.

## 5. Conclusions

The ion precipitation from solutions onto hard surfaces and the formation of grains of sodium chloride are multifactorial mechanisms, which are still hypothetic [51]. Based on the results, it can be concluded that:(1)With increasing film thickness, the root-mean-square roughness R_q_ indices reduces with 56 nm (for bare Ti-6Al-4V) to 36 nm (at a-C:H:SiO_x_ film thickness 3 µm);(2)According to the Rockwell hardness test, the a-C:H:SiO_x_ film with a thickness of 0.5 to 3 μm deposited onto the Ti-6Al-4V alloy surface, had a high adhesive strength (not worse than HF3);(3)The obtained coating was rather durable, since after the five-weak in vitro biodegradation in a 0.9% NaCl solution, the weight of a-C:H:SiO_x_-coated substrates did not change significantly (0.02–0.03% higher compared to initial level before immersion in a solvent);(4)According to the EDX analysis, a-C:H:SiO_x_ films of 0.5 to 3 µm thickness statistically decreased Na^+^ (by 97–99.5%) and Cl^−^ (by 96–98.5%) precipitation on the substrate surface;(5)The multiple regression equations showed that the lower precipitation of Na^+^ and Cl^−^ with 0.99 determination coefficient onto the a-C:H:SiO_x_ film surface, was provided by the growth in the C and Si content in the surface layer;(6)Computer-assisted morphometry of the a-C:H:SiO_x_-coated substrate surface showed a five–twenty-fold reduction in the area occupied by the precipitated grains as compared to the uncoated substrates.

In terms of the potential mechanisms of this phenomenon, the reduction in the a-C:H:SiO_x_ film surface area occupied by NaCl precipitates could also be induced by changes in their surface roughness and phase composition affecting the electrostatic surface potential.

There is a wide variety of physicochemical (thickness, roughness, *sp*^3^/*sp*^2^ ratio, content of hydrogen and alloying additives, wettability, surface free energy, etc.), mechanical, and tribological properties (solidity, adhesive strength, coefficient of friction, wear rate, etc.) of DLC films. Their properties are not always completely controlled, which is caused by the use of different types of substrates and peculiarities of methods of film formation [49]. At present, a shift in emphasis from scattered experiments with DLC films to applied technologies and studying the behavior of specific coatings for each type of implant substrate is apparently needed.

The problem of stable behavior of the coating on cardiovascular stents and friction components of mechanical heart pumps has not yet been solved. For example, clinical trials of DLC coatings on the stents showed both positive [52,53] and unsatisfactory results [54,55]. Therefore, from our point of view, physicochemical solutions for the a-C:H:SiO_x_ coating formation on the cardiovascular stents and circulation pumps seem the most difficult and also require additional studies. To date, we have achieved Na^+^ and Cl^−^ controllable poor mineralization of the a-C:H:SiO_x_ films deposited by PACVD technique on Ti-6Al-4V substrate and immersed in steady-state culture conditions. Of course, a 0.9% NaCl solution is only one of the model biological fluids for test trials. Herein, exact fundamental surface features (wettability, surface free energy, charge sign and zeta-potential, phase and elemental composition, etc.) affecting ion deposition onto the a-C:H:SiO_x_ coating in conditions of fluid stream with using other biological media (i.e., simulated body fluid, synthetic nutrient media) and substrates may be prospective aims of in vitro investigations.

## Figures and Tables

**Figure 1 materials-15-04239-f001:**
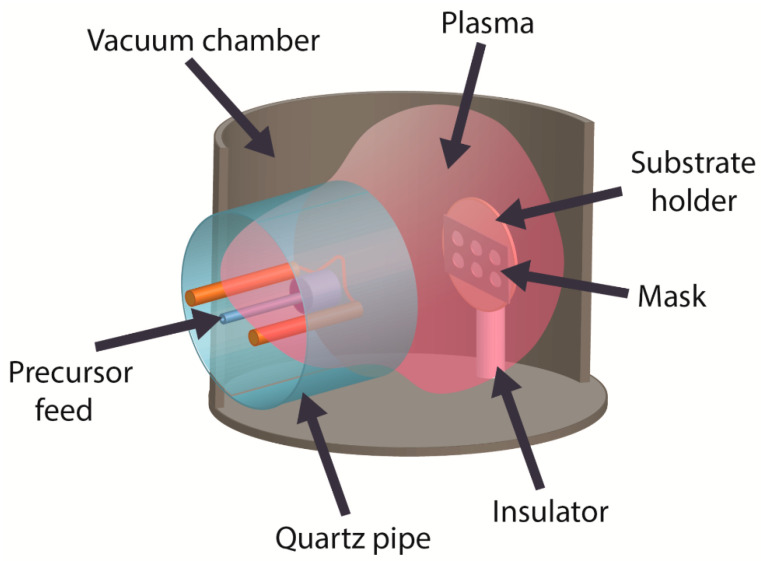
General configuration of the PACVD vacuum system to prepare a-C:H:SiO_x_ films on Ti-6Al-4V substrate.

**Figure 2 materials-15-04239-f002:**
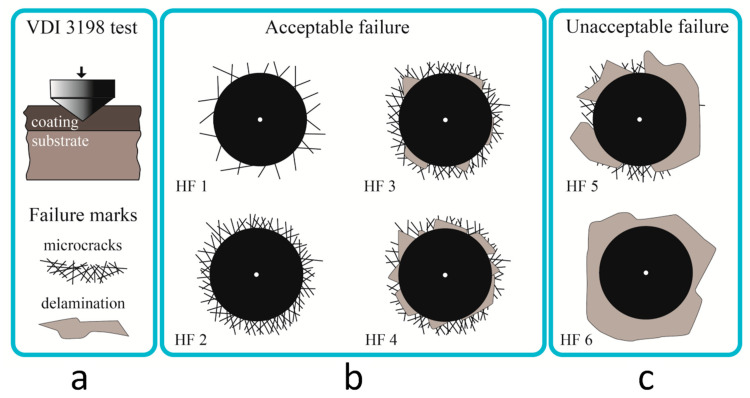
VDI 3198 indentation tests (**a**) and classification of the results as acceptable failure (**b**) and unacceptable failure of the film integrity (**c**).

**Figure 3 materials-15-04239-f003:**
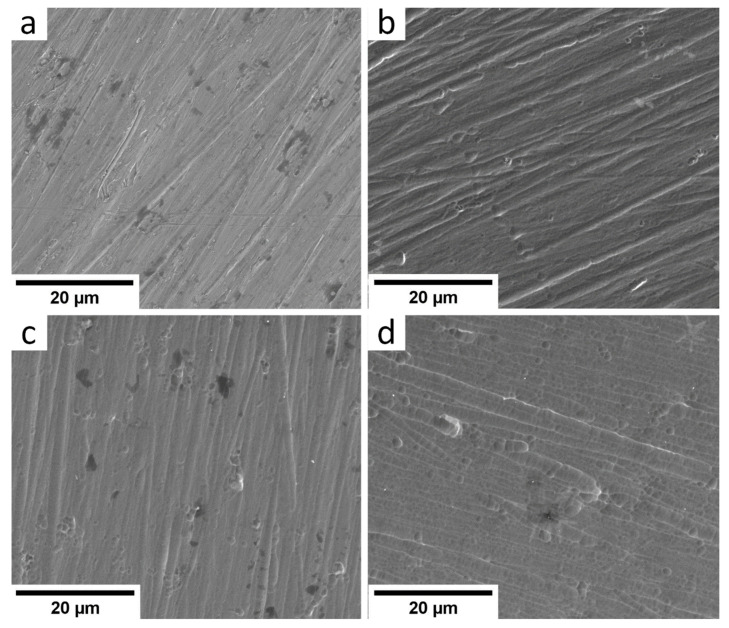
SEM images of a-C:H:SiO_x_-coated Ti-6Al-4V alloy surface before in vitro biodegradation in 0.9% NaCl solution: (**a**) uncoated, (**b**) 0.5 µm thick film, (**c**) 1.5 µm thick film, (**d**) 3 µm thick film.

**Figure 4 materials-15-04239-f004:**
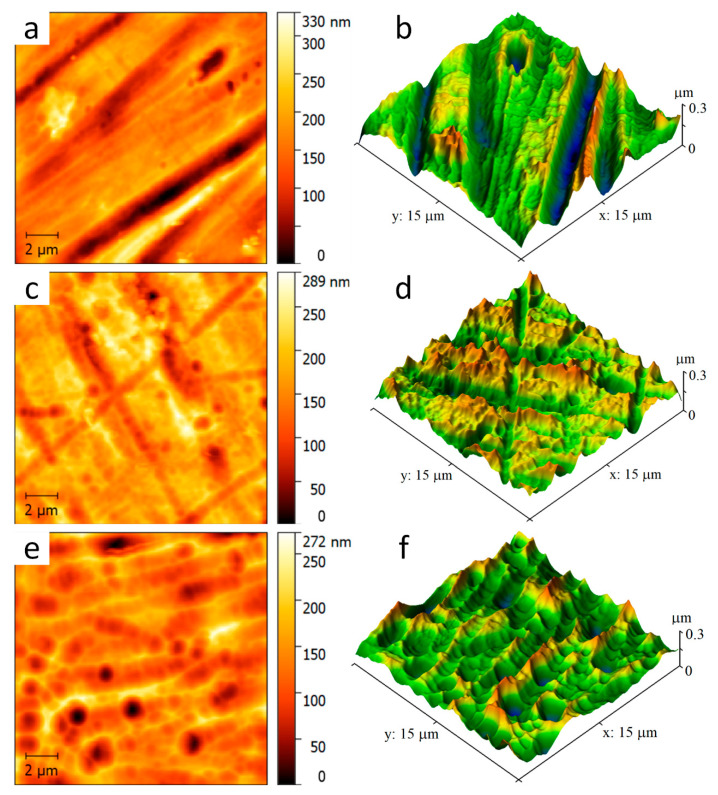
Two- and three-dimensional SPM images of Ti-6Al-4V alloy surface with a-C:H:SiO_x_ film of different thickness before in vitro biodegradation in 0.9% NaCl solution: (**a**,**b**) 0.5 µm, (**c**,**d**) 1.5 µm, (**e**,**f**) 3 µm.

**Figure 5 materials-15-04239-f005:**
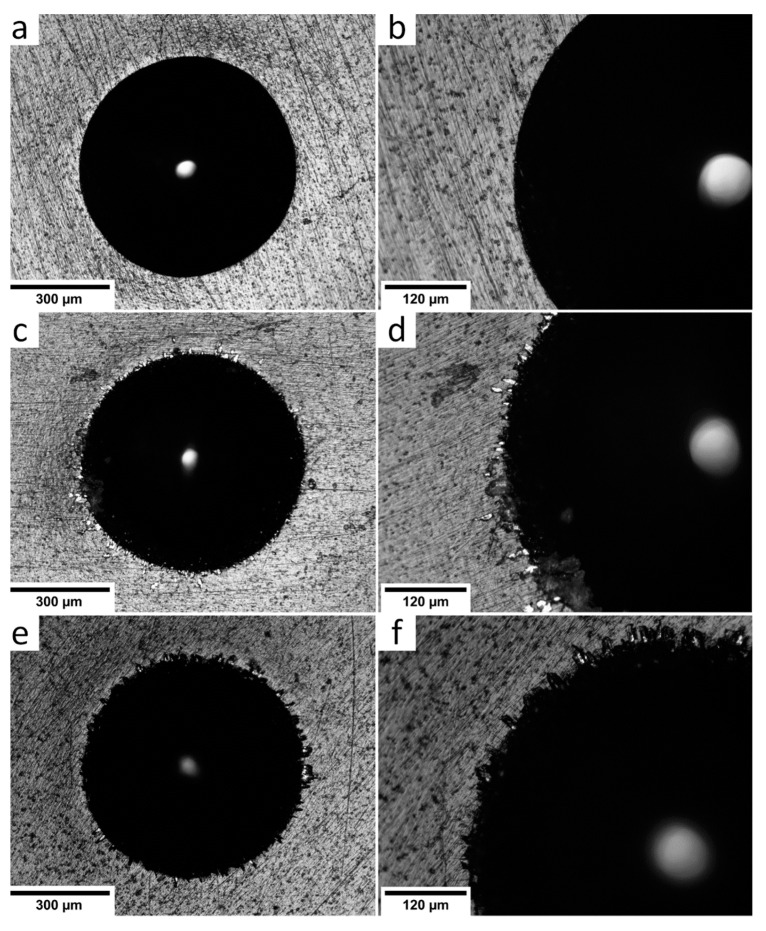
Optical micrographs of Ti-6Al-4V surface with deposited a-C:H:SiO_x_ films of different thickness after the Rockwell indentation process: (**a**,**b**) 0.5 µm, (**c**,**d**) 1.5 µm, (**e**,**f**) 3 µm.

**Figure 6 materials-15-04239-f006:**
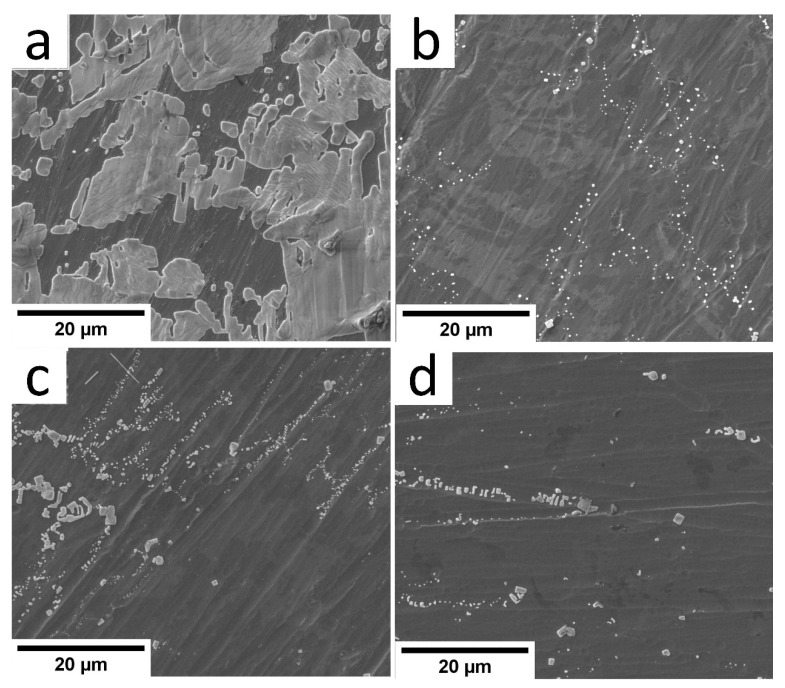
SEM images of a-C:H:SiO_x_-coated substrate surface after five-week in vitro biodegradation in 0.9% NaCl solution: (**a**) uncoated; (**b**) 0.5 µm thick film; (**c**) 1.5 µm thick film; (**d**) 3 µm thick film. Magnification: 5000×.

**Figure 7 materials-15-04239-f007:**
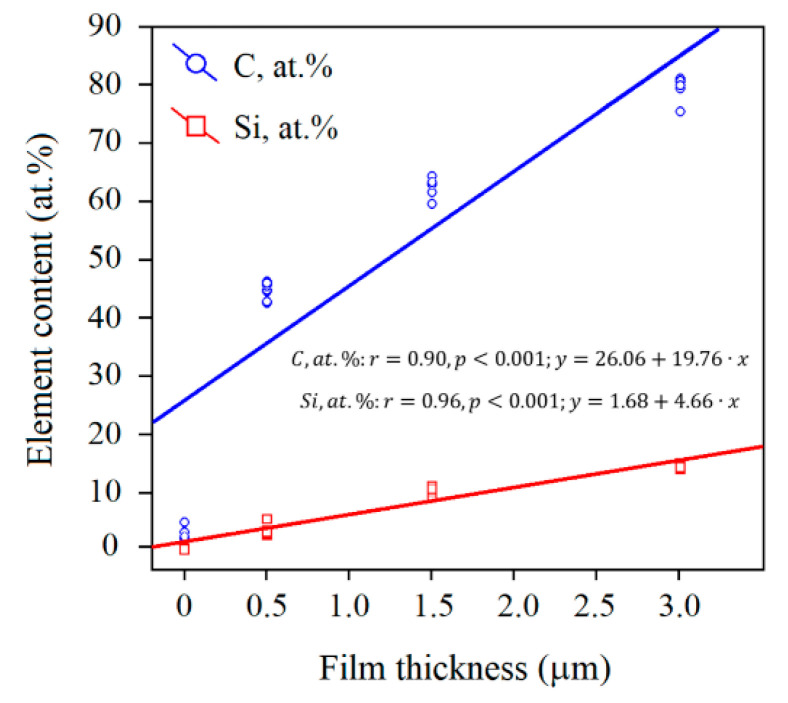
Dual regression dependences of C and Si content in a-C:H:SiO_x_-coated Ti-6Al-4V substrate on the film thickness after five-week immersion in 0.9% NaCl solution. The EDX data are shown on the Y axis.

**Table 1 materials-15-04239-t001:** Elemental composition of Ti-6Al-4V alloy before immersion in 0.9% NaCl solution, X ± SD.

Titanium, wt.%	Aluminum, wt.%	Vanadium, wt.%	Impurities (Fe, Zr, O, C, Si, N), wt.%
88.4 ± 1.2	6.2 ± 0.6	4.1 ± 0.4	˂1.3

**Table 2 materials-15-04239-t002:** Roughness values for Ti-6Al-4V alloy surface with a-C:H:SiO_x_ film of different thickness before in vitro biodegradation in 0.9% NaCl solution, X ± SD.

Thickness, µm	*R*_max_, nm	*R*_mean_, nm	*R*_a_, nm	*R*_q_, nm
Uncoted Ti-6Al-4V	352 ± 22	164 ± 14	45 ± 5	56 ± 7
0.5 ± 0.1	330 ± 20	152 ± 12	39 ± 5	51 ± 7
1.5 ± 0.2	289 ± 16	162 ± 11	29 ± 4	37 ± 5
3 ± 0.3	273 ± 14	129 ± 9	28 ± 4	36 ± 5

*Notation*: *R*_max_—maximum peak height, *R*_mean_—average peak height, *R*_a_—average of a set of individual measurements, *R*_q_—root-mean-square roughness. Each substrate was tested at least at 3 random points.

**Table 3 materials-15-04239-t003:** Ti-6Al-4V substrate weight before and after five-week in vitro biodegradation in 0.9% NaCl solution, *Me* (*Q*1; *Q*3), *n*_1_ = 15.

Substrates	Initial Weight, mg	Dry Weight, Initial Weight Percentage
Uncoated	337.58 (337.46; 342.10)	100.00 (99.99; 100.01)
a-C:H:SiO_x_-coated 0.5-µm thick film	341.62 (340.57; 344.77)	100.02 (100.02; 100.03)
a-C:H:SiO_x_-coated 1.5-µm thick film	343.78 (335.79; 347.15)	100.03 (100.01; 100.04)
a-C:H:SiO_x_-coated 3-µm thick film	341.46 (340.97; 342.30)	100.03 (100.03; 100.05)

**Table 4 materials-15-04239-t004:** a-C:H:SiO_x_ films vs. NaCl precipitation after five-week in vitro immersion, *Me* (*Q*1–*Q*3).

Group Number	Substrates, *n* = 3	Surface Area of NaCl Precipitates, Area Percentage on SEM Image, *n*_1_ = 10
1	Uncoated Ti-6Al-4V	23.23 (13.60–54.88)
2	a-C:H:SiO_x_-coated 0.5 µm thick	1.08 (0.80–1.29), *P*_U1_ < 0.001
3	a-C:H:SiO_x_-coated 1.5 µm thick	4.18 (0.005–6.32), *P*_U1_ < 0.001
4	a-C:H:SiO_x_-coated 3 µm thick	2.06 (0.59–2.81), *P*_U1_ < 0.001

Note: *n*—the number of substrates in each group; *n*_1_—the number of SEM images with measured area in each group.

**Table 5 materials-15-04239-t005:** Elemental (at.%) composition of Ti-6Al-4V substrate surface after five-week in vitro biodegradation in 0.9% NaCl solution, *Me* (*Q*1; *Q*3), *n*=10.

Group Number	Substrates	C	O	Si	Ti	Na	Cl
1	Uncoated Ti-6Al-4V	3.30 (2.65–5.04)	7.08 (6.26–9.45)	0.39 (0.38–0.45)	76.43 (75.25–81.82)	9.63 (2.81–11.98)	4.15 (0.94–5.18)
2	a-C:H:SiO_x_-coated 0.5-µm thick film	45.62 (44.71–5.89) *P_U1_* < 0.001	5.43 (5.27–5.55) *P_U1_* < 0.05	3.27 (3.25–3.41) *P_U1_* < 0.001	45.63 (44.63–46.76) *P_U1_* < 0.001	0.28 (0.25–0.38) *P_U1_* < 0.001	0.07 (0.05–0.15) *P_U1_* < 0.001
3	a-C:H:SiO_x_-coated 1.5-µm thick film	63.29 (62.96–63.45) *P_U1-2_* < 0.001	4.88 (4.38–5.09) *P_U1-2_* < 0.01	10.84 (10.61–11.10) *P_U1-2_* < 0.001	20.72 (20.33–21.34) *P_U1-2_* < 0.001	0.06 (0–0.16) *P_U1_* < 0.001	0.16 (0.13–0.19) *P_U1_* < 0.001
4	a-C:H:SiO_x_-coated 3-µm thick film	80.91 (80.19–81.10) *P_U1-3_* < 0.001	2.98 (2.75–3.09) *P_U1-3_* < 0.001	14.69 (14.49–14.80) *P_U1-3_* < 0.001	1.42 (1.31–1.72) *P_U1-3_* < 0.001	0.06 (0.05–0.07) *P_U1,2_* < 0.001	0.10 (0.09–0.12) *P_U1_* < 0.001

*Note*: *n* is the number of measurements in each substrate group. Significant differences were determined with the Mann–Whitney U test: *P*_U1_*-P*_U3_ < 0.05 vs. the corresponding group number.

**Table 6 materials-15-04239-t006:** Regression coefficients for chemical elements in a-C:H:SiO_x_ film surface after five-week immersion in 0.9% NaCl solution, X ± SD.

Surficial Chemical Elements	Na^+^ Precipitation		Cl^−^ Precipitation	
	Coefficients	*F*-Test, *P* Significance	Coefficients	*F*-Test, *P* Significance
C	−0.632 ± 0.018	<0.001	−0.306 ± 0.012	<0.001
O	−0.791 ± 0.057	<0.001	−0.318 ± 0.037	<0.001
Si	−0.537 ± 0.058	<0.001	−0.301 ± 0.038	<0.001
Ti	−0.594 ± 0.030	<0.001	−0.306 ± 0.020	<0.001

## Data Availability

The data presented in this study are available in the article and Appendix A.

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
