# Peer review of "In Vitro Biodegradation of a-C:H:SiOx Films on Ti-6Al-4V Alloy"

_materials, 2022, doi:10.3390/ma15124239_

Round 1
Reviewer 1 Report
- In the Introduction section, the authors cited the specific results of previous research and cited them adequately. However, they did not mention their shortcomings in previous research. In the Introduction section, the penultimate paragraph should contain common features of previous research. The shortcomings of previous research should also be pointed out, in general.
- In the Introduction section, the last paragraph should contain the scientific contribution and scientific hypotheses of your research. Complete, further elaborate the scientific contribution and scientific hypotheses of your research. Be explicit. In addition to the goal of the research (which was written), the novelty in the context of the scientific contribution should be pointed out. Scientific contributions should be written based on the shortcomings of previous research in the literature. In this way, the authors will better emphasize novelty and scientific soundness.
- Why you choose these alloys?
- Add a table of EDS test for alloys.
- The "2. Materials and Methods" section should not only list materials and methods. Every choice should be explained. Complete this section in detail.
- Analyze and discuss possibilities of practical application.
- Complete the conclusions with the limitations of the proposed methodology. Also write future research.
- Generally, the quality of the writing could be improved.
Author Response
We are grateful to dear Reviewer for the work and generally positive assessment of our paper. Some corrections to the questions and comments of the Reviewer were done in the manuscript by yellow marker.
Point 1: In the Introduction section, the authors cited the specific results of previous research and cited them adequately. However, they did not mention their shortcomings in previous research. In the Introduction section, the penultimate paragraph should contain common features of previous research. The shortcomings of previous research should also be pointed out, in general.
Response 1: We have added some information in the introduction. We pointed out that previous works are mainly focused on the effects of the implanted material on the surrounding tissues and body fluids. This paper, on the other hand, focuses on the effects of body fluid on the coating deposited on the metal substrate.
Point 2: In the Introduction section, the last paragraph should contain the scientific contribution and scientific hypotheses of your research. Complete, further elaborate the scientific contribution and scientific hypotheses of your research. Be explicit. In addition to the goal of the research (which was written), the novelty in the context of the scientific contribution should be pointed out. Scientific contributions should be written based on the shortcomings of previous research in the literature. In this way, the authors will better emphasize novelty and scientific soundness.
Response 2: A novelty of our investigation conditioned by an absence of scientific knowledge about biodegradation of a-C:H:SiOx films with different thickness was initially. We added some contradictions of a behavior of Si-doped coatings in biological fluids to emphasize our scientific contribution to this research field.
Point 3: Why you choose these alloys?
Response 3: Our short explanation about Ti-6Al-4V alloy use was introduced at the beginning of “Introduction” section.
Point 4: Add a table of EDS test for alloys.
Response 4: The table with results of Ti alloy elemental analysis was added to section 2.1.
Point 5: The "2. Materials and Methods" section should not only list materials and methods. Every choice should be explained. Complete this section in detail.
Response 5: The "2. Materials and Methods" section was detailed.
Point 6: Analyze and discuss possibilities of practical application.
Response 6: A discussion of possible practical application was introduced in last paragraph of 4th section.
Point 7: Complete the conclusions with the limitations of the proposed methodology. Also write future research.
Response 7: The conclusions were extended according to reviewer's comments. The additions were marked with yellow color.
Point 8: Generally, the quality of the writing could be improved.
Response 8: We hope our revisions improved the paper's quality.
Reviewer 2 Report
The manuscript of Grenadyorov et al. evaluates the in vitro biodegradation of a-C:H:SiOx films obtained on Ti-6Al-4V alloy. The work is a good fit for the special issue it is submitted to (Future Trends in Chemical Engineering Science: Coatings; Additive Manufacturing, Composites and Inorganic Materials). The following points need to be addressed, as mentioned below:
1. Figure 2: the VDI 3198 test info in the figure should be the first panel. Also denote the panels with a,b,c and provide a more detailed figure caption. Is there also a percentage of delamination attributed to HF4-HF6?
2. Regarding the thickness of the layers of 0.5µm, 1.5 and 3µm, these values were determined in the authors' previous work? If so, authors should include a statement to this in results section.
3. The SPM data, at least in the Table if not the actual images too, should also include the values for the bare alloy.
4. Figure 5 caption - it should clearly state that these images are after the identation process
5. Authors could improve the discussion parts to better emphasize the critical points of the manuscript.
6. Other minor things: a) overall all figure captions should be checked and improved to include all data (e.g. as mentioned above or when it is a correlation from EDX data, etc.); b) if the data in Table S3 contradicts an assumption and contributes to the overall conclusion, it would be better to include it in the main manuscript.
Author Response
We are grateful to dear Reviewer for the work and generally positive assessment of our paper. Some corrections to the questions and comments of the Reviewer were done in the manuscript by blue marker.
Point 1: Figure 2: the VDI 3198 test info in the figure should be the first panel. Also denote the panels with a,b,c and provide a more detailed figure caption. Is there also a percentage of delamination attributed to HF4-HF6?
Response 1: Figure 2 is corrected as recommended. There are techniques that allow determining the peeling coefficient, which is equal to the ratio of the area of peeling of the coating to the area of the hole left by the indenter. When the smaller the value of the peeling coefficient, the adhesion of the coating is the better. This description was added at the end of section 2.4.
A percentage of delamination attributed to HF4-HF6 was not estimated because the film failures did not exceed HF2-HF3 values.
Point 2: Regarding the thickness of the layers of 0.5µm, 1.5 and 3µm, these values were determined in the authors' previous work? If so, authors should include a statement to this in results section.
Response 2: Film thicknesses were currently determined. Corresponded research technique was added at the beginning of section 2.3.
Point 3: The SPM data, at least in the Table if not the actual images too, should also include the values for the bare alloy.
Response 3: Roughness values for the surface of alloy Ti-6Al-4V added to section 3, table 2.
Point 4: Figure 5 caption - it should clearly state that these images are after the identation process.
Response 4: It was done.
Point 5: Authors could improve the discussion parts to better emphasize the critical points of the manuscript.
Response 5: The introduction, discussion and conclusions sections were significantly improved. Insertions were colored by yellow marker.
Point 6: Other minor things: a) overall all figure captions should be checked and improved to include all data (e.g. as mentioned above or when it is a correlation from EDX data, etc.); b) if the data in Table S3 contradicts an assumption and contributes to the overall conclusion, it would be better to include it in the main manuscript.
Response 6: a) All figure and table captions were checked and improved. b) Table S3 was transferred into the main manuscript and renamed as Table 5.
Reviewer 3 Report
The article is about in vitro biodegradation of a-C:H:SiOx films on Ti-6Al-4V alloy. However, some issues must to be addressed:
- Abstract: Please start by expressing the aim of this paper, followed by the rest of the information. Typically, the abstract should provide a broad overview of the entire project, summarize the results, and present the implications of the research or what it adds to its field.
- Please avoid bulk citations like [8-11] …
- Please include an equipment section or insert it in section 2.
- The results are merely presented, not properly discussed. Please add explanations for the observed changes. Please give an extended discussion on the obtained results and correlate your findings with previous literature studies and prospective applications.
- More analysis and interpretation of the results should be added for a clearer understanding of observed experimental phenomena.
- The authors must to provide some details about importance of the research and their applicability.
- Please rewrite the conclusions in a more quantitative form and enhance the clarity of the conclusion section in order to highlight the results obtained.
- General check-up and correction of the English language is suggested. There are still some minor typos and grammatical errors.
The author needs to address the abovementioned points for the betterment of the manuscript.
Author Response
We are grateful to dear Reviewer for the work and generally positive assessment of our paper. Some corrections to the questions and comments of the Reviewer were done in the manuscript by grey marker.
Point 1: Abstract: Please start by expressing the aim of this paper, followed by the rest of the information. Typically, the abstract should provide a broad overview of the entire project, summarize the results, and present the implications of the research or what it adds to its field.
Response 1: Abstract was reconstructed and added significantly by our main results. Future possible application of obtained results was also introduced.
Point 2: Please avoid bulk citations like [8-11] …
Response 2: The references were changed where it was possible. Bulk citations was used to accent a large scale of conducted investigations on the outlined problems
Point 3: Please include an equipment section or insert it in section 2.
Response 3: It seems to us, corresponding equipment is clear presented in each section of Materials and Methods.
Point 4: The results are merely presented, not properly discussed. Please add explanations for the observed changes. Please give an extended discussion on the obtained results and correlate your findings with previous literature studies and prospective applications.
Point 5: More analysis and interpretation of the results should be added for a clearer understanding of observed experimental phenomena.
Response 4 and 5: The questions of a-C:H:SiOx film biodegradation/precipitation is poor discussed in the scientific literature because such results are practically absent. We have emphasized it in Introduction section as our research novelty. We tried to extend our interpretation, discussion and prospective applications in Discussion and Conclusions sections as best we could.
Point 6: The authors must to provide some details about importance of the research and their applicability.
Response 6: Our understanding of importance and possible practical applications of the research was provided with yellow color at the end of "Conclusions" section.
Point 7: Please rewrite the conclusions in a more quantitative form and enhance the clarity of the conclusion section in order to highlight the results obtained.
Response 7: It was done by grey marker.
Point 8: General check-up and correction of the English language is suggested. There are still some minor typos and grammatical errors.
Response 8: According to our correspondence with Editorial Office of Materials journal, English style correction will be provided by Editorial Office if the paper is accepted for publication.
Round 2
Reviewer 1 Report
The presented data are original and interesting. The manuscript has been significantly improved and is suitable for publication in the present Journal.
Reviewer 2 Report
The authors addressed or the points raised by the reviewer and made the corresponding modifications in the manuscript. I would recommend the authors to make a final check for minor corrections (English, mispelling).
Reviewer 3 Report
The article is suitable for publication.